

# Are BVOC exchanges in agricultural ecosystems overestimated? Insights from fluxes measured in a maize field over a whole growing season

A. Bachy[1], M. Aubinet[1], N. Schoon[2], C. Amelynck[2], B. Bodson[3], C. Moureaux[1], B. Heinesch[1]

[1]Department BIOSE, Gembloux Agro-Bio-Tech, University of Liège, Gembloux, 5030, Belgium
[2]Department Mass sprectrometry, Belgian Institute of Space Aeronomy, Uccle, 1180, Belgium
[3] Department Agronomical Sciences, Gembloux Agro-Bio-Tech, University of Liège, Gembloux, 5030, Belgium

*Correspondence to*: B. Heinesch (bernard.heinesch@ulg.ac.be) and A. Bachy (aurelie.bachy@ulg.ac.be)

**Abstract.** Although maize is the second most important crop worldwide, and the most important C4 crop, no study on biogenic volatile organic compounds (BVOCs) has yet been conducted on this crop at ecosystem scale and over a whole growing season. This has led to large uncertainties in cropland BVOC emission estimations. This paper seeks to fill this gap by presenting, for the first time, BVOC fluxes measured in a maize field at ecosystem scale (using the disjunct eddy covariance by mass scanning technique) over a whole growing season in Belgium. The maize field emitted mainly methanol, although exchanges were bi-directional. The second most exchanged compound was acetic acid, which was taken up mainly in the growing season. Bi-directional exchanges of acetaldehyde, acetone and other oxygenated VOCs also occurred, whereas the terpenes, benzene and toluene exchanges were small, albeit significant. Surprsingly, BVOC exchanges were as same order of magnitude on bare soil and on well developed vegetation, suggesting that soil is a major BVOC reservoir in agricultural ecosystems. Quantitatively, the maize BVOC emissions observed were lower than those reported in other maize, crops and grasses studies. The standard emission factors (SEFs) estimated in this study ($231 \pm 19$ µgm$^{-2}$h$^{-1}$ for methanol, $8 \pm 5$ µgm$^{-2}$h$^{-1}$ for isoprene and $4 \pm 6$ µgm$^{-2}$h$^{-1}$ for monoterpenes) were also much lower than those currently used by models for C4 crops, particularly for terpenes. These results suggest that maize fields are small BVOC exchangers in north-western Europe, with a lower BVOC emission impact than that modelled for growing C4 crops in this in this part of the world. They also reveal the high variability in BVOC exchanges across world regions for maize and suggest that SEFs should be estimated for each region separately.

## 1 Introduction

In order to model future climate with high reliability, an in-depth understanding of all climate components and their interactions is necessary. Volatile organic compounds (VOCs) are among these components. Although VOCs constitute only a small fraction of the total air composition, their high reactivity has a significant affect on atmospheric chemistry and climate by affecting the methane lifetime in the atmosphere (Isaksen et al., 2009; Williams et al., 2013) and through





formation of secondary organic aerosols (Sartelet et al., 2012; Ziemann and Atkinson, 2012) and tropospheric ozone (Fry et al., 2012; Isaksen et al., 2009; Sartelet et al., 2012; Tsimpidi et al., 2012). The understanding of VOC exchanges is therefore a research priority if better climate and air quality predictions are to be achieved (Lerdau and Slobodkin, 2002; Osborne et al., 2010).

There are numerous VOC sources (e.g., solvents, burning residues, micro-organisms). There is however a general consensus that most atmospheric VOCs originate from biogenic sources (hence the term biogenic VOCs, BVOCs), and particularly from plants (Fowler et al., 2009).

BVOC exchange composition and dependence on environmental factors are plant species-specific (Monson et al., 2013), and BVOC studies therefore need to broaden the range of investigated plants and ecosystems in order to estimate global BVOC

exchanges with more accuracy (Lerdau and Slobodkin, 2002; Niinemets et al., 2014). This is currently not the case, however. Although forests have been the most widely studied ecosystem (see Niinemets et al., 2013 for a review), only few BVOC studies have focused on croplands (Copeland et al., 2012; Crespo et al., 2013; Eller et al., 2011; Karl et al., 2005; Konig et al., 1995; Warneke et al., 2002). So far as we know, only two BVOC measurement studies have dealt with maize (Das et al., 2003; Graus et al., 2013), although it is the second most important crop in the world in terms of cultivated area

(FAOSTATS), and in some regions is the dominant crop (e.g., the Corn Belt zone in the USA), and it could become even more important in meeting growing food needs (Hardacre et al., 2013) and enhancing biofuel production (Bellarby et al., 2010). In addition, both maize studies were conducted over only a few days and under poorly contrasted weather conditions, and were thus unable to evaluate the relative effects of climate and phenology on BVOC exchanges. Knowledge about BVOC exchanges from maize therefore remains very limited.

In order to fill this scientific gap, this study focused on BVOCs exchanged by maize, based on ecosystem-scale measurements performed over a whole growing season. So far as we know, it is the first study dealing with long-term BVOC measurements on maize. It sought to answer the following questions:

- Which BVOCs are exchanged in a maize field?

- How important is growing maize in terms of BVOC exchanges compared with other agricultural crops?

- What quantity of BVOCs is exchanged in a maize field under standard environmental conditions?



## 2 Materials and Methods

### 2.1 Experimental set-up

#### 2.1.1 Site and variety

The study was carried out on a silage maize (*Zea mays L.*, varieties Prosil and Rocket) field about 185 x 255 m at the Lonzée

Terrestrial Observatory (LTO) in Belgium (50°33'08" N, 4°44'42" E) from 17 May to 11 October 2012. The field is on a
plateau with a small slope of 1.2%. There is more information about the LTO site in Moureaux et al. (2006).

The maize was sown on 17 May, emerged on 25 May and was harvested by 13 October 2012. Fertilizers were applied on 4
June 2012, and no measurements were taken between 30 May and 8 June 2012 in order to prevent fertilizer pollution of the
instruments. During the study, the field was surrounded by sugar beet and there was a silo about 300 m north-west of the

measurement mast. Subsequent analysis did not detect any significant influence of the surrounding crop or silo on the
measured fluxes.

#### 2.1.2 Flux measurements

The BVOC fluxes were computed every half-hour from high frequency vertical wind speed and BVOC concentration
measurements using the disjunct eddy covariance by mass scanning (DEC-MS) technique. In this paper, fluxes are expressed

15  per m² of soil and a positive flux indicates an emission from the ecosystem to the atmosphere.

The vertical wind velocity was measured at 20 Hz with a 3D sonic anemometer (Solent Research R3, Gill Instruments
Lymington, UK) mounted on a 2.7 m-high mast. Due to maize growth and in order to have a reasonably aerodynamic
measurement height, the anemometer was raised to 3.5 m high from 12 July 2012 to 17 August 2012 and to 3.9 m high from
that date to harvest.

The BVOC concentrations were measured with a high sensitivity proton-transfer-reaction mass spectrometry (hs-PTR-MS)
model (Ionicon Analytick GmbH, Innsbruck, Austria). Ambient air was continuously sampled close to the sonic anemometer
through a main sampling line in perfluoroalkoxy alkanes (PFA) (Fluortechnik-Wolf, Esslingen-Berkheim, Germany), 18/20
m long (before and after 12 July 2012, respectively) and with an inner diameter of 6.4 mm and a flow rate ranging from 13 to
13.5 STP L min$^{-1}$ (Standard Temperature and Pressure conditions corresponded to 273.15 K and 101.3. kPa). In order to

prevent water vapour condensation in the main sampling line, which could dissolve some BVOC compounds, the sampling
line was thermally insulated and heated a few degrees above ambient temperature. A polytetrafluoroethylene (PTFE) filter
(Pall, 47 mm diameter, 2 micron pore size) was installed 2/4 m (before and after 12 July 2012, respectively) downstream of
the main sampling line inlet in order to keep the tube clean. Part of the air flow (0.1 STP L min$^{-1}$) was drawn into the hs-
PTR-MS through a 1 m-long heated peek capillary with an inner diameter of 1 mm. The hs-PTR-MS and the sub-sampling

line were installed in a temperature-controlled shelter (293 K) 15 m from the measurement tower. Since no significant
impact of the shelter on trace gas fluxes was identified, it was assumed that the distance between the shelter and





measurement tower was enough to prevent wind distortion by the shelter from having a significant impact on wind conditions near the measurement tower.

### 2.1.3 Ancillary measurements

$H_2O$ fluxes, friction velocity $u_*$ and micro-meteorological variables were measured together with BVOC fluxes at a half-hourly scale. $H_2O$ fluxes were measured using the conventional eddy covariance technique, with the same sonic anemometer as for BVOCs and a closed-path infrared gas analyzer (IRGA, Li-7000, LI-COR, Lincoln, NE, USA). $u_*$ was measured with the sonic anemometer. Monitored meteorological variables relevant to this study were photosynthetic photon flux density (PPFD) (BF3, Delta_T Devices Ltd, Cambridge, UK) and air temperature (T) (RH T2, Delta_T Devices Ltd, Cambridge, UK). There is more information about the non-BVOC experimental set-up in Aubinet et al. (2009).

The phenological development of the maize was followed up through weekly site visits and from pictures taken every day with a phenological camera. Biomass and leaf area index (LAI) were measured on 7 and 4 dates, respectively, between the end of June and the harvest. The evolutions of the biomass and the LAI during the maize growing season were then evaluated by fitting a sigmoid function to the data. In addition, biomass and LAI were set at zero before maize emergence and set at their maximal value after 20 August 2012, in line with field observations.

### 2.2 PTR-MS operation

The hs-PTR-MS instrument was operated in the multiple ion detection mode. Thirteen different ion species, of which 11 were related to BVOCs (Table 1), were measured in a single measurement cycle. The dwell time for each ion species was 0.2 s and the total cycle time was 3.25 s.

The hs-PTR-MS was operated with a drift tube pressure of 2.1 hPa, a drift tube temperature of 333 K and a drift voltage of 600 V. The instrumental background was determined for 20 min every 4 h by switching the hs-PTR-MS inlet flow from ambient air to BVOC-free air, which was obtained by sending ambient air through a heated catalytic converter. Only the final 8 min were used for calculating the mean background values. The sensitivity of the instrument was calibrated every 2-3 days using a gravimetrically prepared mixture of BVOC gases in $N_2$ (Apel-Riemer Environmental, Denver, CO, USA). The initial mixture, containing about 1 ppmv methanol, acetaldehyde and acetone and about 0.5 ppmv acetonitrile, isoprene, methyl vinyl ketone (MVK), methacrolein (MACR), methyl ethyl ketone (MEK), α-pinene and sabinene, with an accuracy of 5%, was used until 22 May 2012. From that date on, a second mixture containing about 1 ppmv methanol, acetaldehyde, MEK and cis-3-hexenol and about 0.5 ppmv acetonitrile, acetone, isoprene, MACR, MVK, benzene, toluene, m-xylene, α-pinene and sabinene, with an accuracy of 5%, was used instead. The compounds were further diluted (2-12 ppbv range) using a dynamic dilution system. Additionally, three calibrations as a function of relative humidity were performed during the study.

In this study, it was assumed that only acetone and acetic acid contributed to the m/z 59 and 61 ion signals, respectively. The calibration factor for acetic acid was estimated from the experimentally determined one for acetone by taking into account





the fragmentation of the protonated molecules in the drift tube (Inomata and Tanimoto, 2010; Schwarz et al., 2009) and the ratio of the calculated collision rate constants (Su, 1994) and by assuming the same transmission efficiency for ions at m/z 59 and 61.

The m/z 69 signal (M69) is commonly associated to protonated isoprene in hs-PTR-MS BVOC studies, but it may also result from dissociative proton transfer to other BVOCs (Table 1). M69 emissions were observed on bare soil (Fig.1, the soil is bare during the stage G as explained in Table 2), but they did not significantly increase with T or PPFD (data not shown), indicating that they probably did not originate from isoprene sources during that period, as isoprene emissions are known to be strongly influenced by those driving parameters (Niinemets et al., 2013). However, when the crop developed, M69 emissions did increase with T and PPFD (data not shown) and could therefore, at least partly, be due to isoprene. Since our experimental set-up did not allow unambiguous identification of the compounds resulting in M69 fluxes, the calibration factor for isoprene was used in the M69 flux calculation. The M69 fluxes were therefore considered as an upper isoprene flux limit for all phenological stages apart from G.

The ion signal at m/z 83 was measured in order to represent green leaf volatiles (GLV, Table 1) and to evaluate the maize stress status. The observed exchanges were qualitative, however, and therefore this signal is not discussed in this paper.

In order to compare the water vapour fluxes obtained with the hs-PTR-MS and the IRGA, the hs-PTR-MS ion signal at m/z 39 (M39), a water vapour proxy, was post-calibrated using the mean half-hourly $H_2O$ concentration measured by the IRGA ($R^2$=0.97), as described by Ammann et al. (2006).

## 2.3 BVOC flux computation

The general BVOC flux computation framework drew on the EUROFLUX methodology (Aubinet et al., 1999), which is designed for $CO_2$ flux measurements using the conventional EC technique. It was adapted for DEC-MS and low flux signal-to-noise ratio specificities when relevant. Means were computed using block averaging over 30 min periods and a 2D rotation was applied. The time lag between the wind and the concentration data streams was calculated using the technique recommended by Bamberger et al. (2010), Hörtnagl et al. (2010) and Taipale et al. (2010), whereby time lags and fluxes are determined by covariance maximisation, using the disjunct concentrations (so without imputation) to compute the cross-correlation curve and applying a smoothing function (a 3 s window size was chosen) to that curve prior to peak determination. The time lag mode found for methanol was 5.25 s, close to the M39 time lag mode (5.45 s) and to the theoretical time lag determined in situ by isopropylalcohol injection (5 ± 0.5 s). No mode was found for the other compounds investigated. A default value of 5.25 s was used when the maximal covariance was not found within the [2.75-7.75] s time window (between 49 and 78% of the data depending on the compound).

A frequency response correction accounting for low-pass filtering was applied on eddy flux data. The approach described by Moncrieff et al. (1997) was used, whereby each instrumental effect is modelled using a theoretical transfer function. The total transfer function was characterized by a half-power cut-off frequency of 0.4 Hz and was combined with the theoretical





Kaimal co-spectrum, which is very close to the experimental sensible heat co-spectra (Kaimal and Finnigan, 1994), in order to determine a correction factor to apply to the fluxes. This factor ranged between 1 and 2, with a mean value of 1.29.

## 2.4 BVOC flux filtering

BVOC fluxes were discarded when the maize field contribution to the total flux footprint (Neftel et al., 2008) was below 70% (2% of the whole dataset), during hs-PTR-MS calibration and background measurement periods (22%), when the flux measurement system was stopped because of maintenance operations (1%), power failures (9%) or spraying events (6%), and when the fluxes were not computed because concentration or sonic anemometer data were not available (9%). This resulted in 3592 valid half-hourly flux data for each investigated compound (51% of the whole dataset).

It should be noted that we did not filter BVOC fluxes according to a lower $u_*$ threshold or to stationarity. Indeed, $u_*$ can actually control soluble BVOC fluxes (Aubinet, 2012; Laffineur et al., 2012). Moreover we observed that $u_*$ induced a higher flux random error, but not a systematic error. For all the investigated compounds, both daytime and nighttime fluxes presented a conical shape when plotted against $u_*$ with the flux range increasing along with $u_*$. The flux detection limit (Sect. 2.5.2) was also positively well correlated with $u_*$, with $R^2$ ranging from 0.51 to 0.80, depending on the compound. In addition, the correlation between the $H_2O$ fluxes measured with the IRGA and those measured with the hs-PTR-MS (M39) improved with decreasing $u_*$. Even if the random error increased with $u_*$, however, the mean exchanges of non-soluble compounds, evaluated by averaging fluxes per $u_*$ class, did not significantly differ with $u_*$. This means that low $u_*$ values did not create biased BVOC fluxes. Consequently, we did not apply a low $u_*$ threshold to those fluxes.

The stationarity criteria designed for the conventional EC technique (Aubinet, 2012) and commonly used in BVOC flux studies were irrelevant for the BVOC fluxes measured at LTO. Stationarity filtering criteria calculated from M39 data (using the approach described by Foken and Wichura, 1996) did not remove the same data as those calculated from the IRGA $H_2O$ data. In addition, for all the investigated BVOC compounds, both stationary and non-stationary data had similar diurnal dynamics and were correlated with the same environmental variables, suggesting that non-stationary data were not abnormal. Therefore, we did not apply that filtering criterion to the BVOC fluxes.

## 2.5 BVOC flux error evaluation

### 2.5.1 Systematic error

The possible occurrence of a systematic error in the calculated BVOC fluxes was evaluated by (i) computing the flux distribution for each compound and (ii) comparing the $H_2O$ turbulent fluxes computed from M39 with those computed from the IRGA data, following the approach used by Ammann et al. (2006).

The half-hourly flux distribution was quite symmetric around zero for each compound. No mirroring effect (Langford et al., 2015) was observed, indicating that the chosen time lag method did not create bias in the flux. Moreover, the M39 fluxes correlated well with the $H_2O$ fluxes measured with the IRGA, even though the determination coefficient was lower



($R^2$=0.71) than the one reported by Ammann et al. (2006) ($R^2$=0.92). The regression slope did not significantly differ from 1, indicating that the $H_2O$ fluxes calculated with the DEC-MS technique were not biased.

Consequently, we considered that the BVOC fluxes measured at LTO were not biased.

### 2.5.2 Random error

An individual 30-min flux random error was equated with its detection limit. The latter was computed as the standard deviation $\sigma$ of the covariance function on a time lag window far from the theoretical time lag and therefore physically irrelevant, following the approach used by Spirig et al. (2005). Depending on the compound, 63 to 86% of the flux data were above the detection limit (moving to 15 and to 60% when $3\sigma$ was taken, as proposed by some authors). Langford et al. (2015) found that the root-mean-square (RMS) of the covariance function was a better estimator of flux random error than $\sigma$. When the RMS was compared with $\sigma$ for fluxes measured in June 2012, however, we did not observe any significant differences (data not shown). The standard deviation method was therefore retained.

Although many BVOC fluxes were lying above their detection limit, flux data performed at the half-hourly scale were very scattered (except for methanol flux). Thus, in all analyses, fluxes were averaged over many observations during a specific period, typically the whole growing season or phenological stage.

### 2.6 BVOC budget computation

The BVOC budget was computed over the whole maize growing season for each compound separately, using the method recommended by Bamberger et al. (2014), with gaps smaller than or equal to 2 h (23% of the whole dataset) filled by linear interpolation, and gaps larger than 2 h (26%) filled using the mean diurnal variation (MDV) technique, with a 16-day window size centered on the missing data.

The budget error was evaluated by flux error propagation. The flux detection limit ($1*\sigma$) was used as the flux random error for measured fluxes. As gap-filled fluxes were determined from measured fluxes, the random error of each individual gap-filled flux was evaluated by propagating the error of the fluxes used to estimate that flux.

The error caused by the gap-filling technique itself was not quantified, but we argue that budgets reported in this paper are consistent for their order of magnitude. First, the gap-filling did not change the BVOC flux pattern. Then, when estimating flux data for gaps smaller than 2h, the use of interpolation techniques other than linear resulted in flux values that were not significantly different from those obtained with linear interpolation. Thirdly, the MDV technique which was used to estimate fluxes in larger gaps has been shown to result in less error than other gap-filling techniques (Bamberger et al., 2014). In addition, most gaps lasted less than 1 day. 8 gaps exceeded 1 day; four of them lasted less than 2 days, and three of them lasted between 3 and 4 days. Only one gap lasted 10 days because of a spraying event. The uncertainty induced in the budget when estimating missing flux data for that gap was evaluated by filling all missing data either with the lowest or with the highest flux values measured within one month window size around the missing data. These extreme flux data induced a variation up to 257, 51, 39 and 53 $g_{BVOC}ha^{-1}$ in the methanol, acetaldehyde, acetone and acetic acid budgets, respectively.





Although the budget was significantly modified (Table 5), its order of magnitude remained similar. Moreover, filling all missing data with extremely low or high flux is realistic only if weather conditions were particularly warm and dry or wet and cold when the gap occurred and if they remained constant during the whole gap duration, which was not the case at LTO. Indeed, a warm and dry period occurred during the first 5 days of the gap and was followed by a wetter and colder

period during the 5 other days. Consequently, the BVOC budget that actually occurred during that period was certainly less extreme than the BVOC budget estimated by considering extreme flux values and therefore closer to the budget estimated using the MDV technique.

## 3 Results and discussion

### 3.1 Maize phenological development

The maize growing season was divided into five distinct phenological periods: G (germination – BBCH 00 to 14); L (leaf unfolding – BBCH 14 to 16); S (stem elongation and leaf area development – BBCH 30 to 39); R1 (inflorescence development, flowering and grain emergence – BBCH 51 to 71); and R2 (grain maturation – BBCH 71 to 89). All the stages were determined by visual observations of the maize field, with the exception of the pivotal date between the stages R1 and R2, which was based on the difference in daily biomass growth because the visual observations were not good enough

determine where R1 ended and R2 began.

A detailed description of all the stages and their correspondence with BBCH codification (Meier, 2001), which is commonly used for crop phenological description, is given in Table 2. Briefly, the ecosystem consisted of bare soil in stage G. The maize developed during the other stages, and so the ecosystem then included both soil and plants. Vegetative growth (i.e., leaves and stem) occurred during stages L and S. Reproductive growth (i.e., flowers and grains) occurred during stages R1

and R2. Vegetative growth was small (R1) to negligible (R2) during the reproductive growth period. It should be noted that, as the maize variety grown at LTO is a 'stay green' variety, it was harvested before entering in senescence, and therefore the senescence phase was not included in the phenological description.

### 3.2 Representativity of BVOC exchanges measured at LTO

The maize variety grown at LTO was intended for silage (livestock feed) production purposes and the management practices

commonly used in this region for this type of crop were thus applied. The weather conditions during the study were among normal for the time and place (Royal Meteorological Institute, Belgium).

The BVOC composition, flux range and budget presented in this study are therefore representative of the fields of maize grown under natural weather conditions in the Hesbaye region of Belgium and, by extension, in north-western Europe, where maize is grown under similar pedo-climatic conditions (i.e., temperate maritime climate and silt or sandy-loamy soils) and

for similar production purposes (i.e., farms with crops and livestock).





## 3.3 BVOCs exchanged at LTO

Throughout the study, methanol was the main compound exchanged (Fig.1), ranging from 31 to 76% of the total mean BVOC exchanges (Table 3, all percentages are given in mass basis). Methanol emissions were observed for all stages apart from L, which was characterized instead by uptakes resulting from wetter and colder conditions (data not shown).

Apart from methanol, other oxygenated VOCs (OVOCs; in this paper, this include methanol, acetic acid, acetaldehyde, acetone, MVK+MACR and MEK) were exchanged. The acetic acid, acetaldehyde, acetone, MVK+MACR and MEK contributions to the total BVOC exchange ranged from 0 to 22% during the phenological stages. Acetic acid, in particular, was the second most important compound exchanged over the growing season, contributing up to 16% of the total exchange for a single phenological period. It was taken up by the ecosystem throughout the growing season, apart from some days

during stage R1 which were characterized by warm and dry conditions and during which small but significant acetic acid emissions were observed instead (Fig.2). Acetaldehyde and acetone fluxes were important during phenological stages, with contribution up to 22% for acetaldehyde and 7% for acetone, but their exchanges varied in magnitude and direction among the phenological stages, resulting in a small acetaldehyde uptake (5%) and a non-significant acetone exchange over the whole growing season. Small but significant MVK+MACR uptake occurred over the whole growing season, accounting for

up to 4% of the total BVOC exchange. Uptake was more pronounced in stage R1, probably due to higher mixing ratios during that period (up to 1.2 ppbv as opposed to 0-0.4 ppvb for other stages), which favored dry deposition mechanisms (Niinemets et al., 2014; Tani et al., 2010). MEK was emitted from stages L to R1 and was taken up during the other stages. MEK exchanges were always significant, but never exceeded 5% of the total BVOC exchange.

Terpenes exchanges (in this paper, terpenes include monoterpenes and isoprene, the maximal exchange rate for the latter

being estimated from M69 fluxes for all stages apart from G, Sect. 2.2) were 1 order of magnitude smaller than methanol exchanges and contributed up to 9% to the total BVOC exchange for a single compound. Significant emissions were found for both compounds during all stages, apart from R2, during which uptake of monoterpenes was observed instead.

Small but significant benzene and toluene uptake was observed for all phenological stages. Each compound contributed up to 7% to the total BVOC exchange.

Besides, each investigated BVOC showed different seasonal dynamics, indicating that the sources, sinks strength and/or exchange mechanisms differed for each compound.

The BVOC exchange composition observed at LTO matched those observed on diverse croplands and grasslands fairly well. The preponderance of methanol emissions over all other BVOCs has been reported in numerous cropland and grassland studies (Bamberger et al., 2010; Copeland et al., 2012; Crespo et al., 2013; Custer and Schade, 2007; Eller et al., 2011;

Ruuskanen et al., 2011; Warneke et al., 2002), including maize studies (Das et al., 2003; Graus et al., 2013). Smaller (compared with methanol exchanges) but significant bi-directional exchanges of other OVOCs and terpenes were also reported in those studies.





The maize field at LTO was not an important monoterpene source, in contrast to the observation reported from an American maize field (Das et al., 2003). It was also a small toluene and benzene sink, whereas both compounds were found to be emitted by maize leaves in another study (Graus et al., 2013). In addition, our observations disagree with the hypothesis proposed by White et al. (2009) that maize could be an important toluene source.

## 3.4 Role of soil in BVOC exchanges at LTO

The soil played an important role in the BVOC exchanges at LTO. Bare soil (stage G) showed emissions of methanol, acetaldehyde and acetone, and the strongest acetic acid uptake occurred during this stage.

It was reported by Schade and Custer (2004) that agricultural soils emit methanol and acetone under warm conditions. The maximal methanol (335 $\mu$gm$^{-2}$h$^{-1}$), acetone (136 $\mu$gm$^{-2}$h$^{-1}$) and acetaldehyde (102 $\mu$gm$^{-2}$h$^{-1}$) emissions recorded at our site in stage G were smaller than maximal emission values found by these authors (Table 4), but they were all within the range of the maximal emissions reported in the review of Peñuelas et al. (2014), i.e., 3-553 $\mu$gm$^{-2}$h$^{-1}$ for methanol, 4-806 $\mu$gm$^{-2}$h$^{-1}$ for acetone and 1.7-102 $\mu$gm$^{-2}$h$^{-1}$ for acetaldehyde.

Interestingly, however, the methanol and acetaldehyde fluxes measured at our site were of the same order of magnitude for bare soil as for fully developed vegetation (R1), both stages occurring under similar weather conditions, and the highest acetone emissions occurred during stage G. This means that soil BVOC exchanges were as important, if not more so, as plant BVOC exchanges at LTO. This observation goes against the current assumption that plant BVOC exchanges dominate soil BVOC exchanges (Peñuelas et al., 2014), at least for our ecosystem.

It has been shown that maize leaves emit methanol, acetone and acetaldehyde (Graus et al., 2013). Moreover, significant methanol, acetone and acetaldehyde emissions have been measured in controlled chambers from maize leaves of the 'Prosil' variety (Mozzaffar, 2015 comm. pers), which is one of the two varieties grown at LTO. At ecosystem-scale, BVOC exchanged by maize should therefore increase along with plant development and associated biomass increases, and lead to higher methanol, acetone and acetaldehyde emissions during stage R1, compared with stage G, but this was not the case at LTO. We therefore conclude that, at least for those compounds, the soil source strength decreased during the maize growing season at LTO and thus reduced the importance of the soil in the total net measured BVOC exchange.

## 3.5 Importance of maize as a BVOC exchanger compared with other crops

### 3.5.1 Comparison of BVOCs exchanged at LTO with other maize BVOC studies

So far as we know, there have been only two other BVOC studies focusing on maize. They were conducted in the United States of America over a few days and under particular weather conditions (Table 4).

We averaged the BVOC fluxes gathered at LTO under similar T, PPFD and phenological stages as the American studies in order to exclude as far as possible the phenological and meteorological effects on BVOC exchanges (Fig.2). The results



showed that BVOC fluxes were much lower at LTO than in the two other maize studies, differing by a factor of 3 to 43 compared to Graus et al. (2013), depending on the compound, and by 2 orders of magnitude compared to Das et al. (2003). The mean exchange reported by Graus et al. (2013), however, arose from leaf-scale measurements performed in controlled chambers. They were therefore affected by up-scaling issues and probably differed from the mean exchanges that would

have been measured at their site at ecosystem scale and under natural environmental conditions. It is possible, therefore, that the discrepancies with our observations result from differences in experimental design.

In contrast, the BVOC flux study by Das et al., 2003 was conducted at ecosystem-scale under natural conditions. Their flux measurement technique (gradient) differed from the one we used (DEC-MS), but it has been shown in other BVOC flux measurement studies that both techniques lead to similar BVOC exchange magnitudes (Park et al., 2014; Karl et al., 2001).

The two orders of magnitude difference in BVOC exchanges between their study and ours therefore reflects real and significant differences in BVOC exchanges between the two sites.

In this comparison, the BVOC exchanges have been normalized by T, PPFD and phenology. Other environmental conditions (e.g., humidity, soil type, soil fertility), as well as cultural regional practices (e.g., chosen maize variety, cultural management such as fertilizer use, irrigation), could, however, directly influence the BVOC exchange magnitude through

constitutive or stress-induced pathways. They could also affect maize growth and phenology and therefore indirectly influence BVOC exchanges. Given the huge differences in normalized BVOC exchange rates among studies, we conclude that some of these other environmental conditions play a major role in the amount of BVOCs exchanged in a maize field, and therefore that BVOC exchange rates obtained for maize in one region cannot be extrapolated to another region simply by normalizing T and PPFD. We also conclude that the importance of maize fields in BVOC exchanges varies strongly among

regions of the world and therefore, in the next section, we will compare the BVOC exchanges measured at LTO only with those reported for other crops grown in Europe.

### 3.5.2 Comparison of BVOCs exchanged at LTO with other crops

The BVOCs exchanged at LTO were compared with diverse C4 and C3 crops and with mixed grassland species. Grasslands were included because their BVOC exchange composition is qualitatively similar to that of crop species (Bamberger et al.,

2010; Crespo et al., 2013; Graus et al., 2013; Custer and Schade, 2007; Eller et al., 2011). It should be noted that, as this study focused on BVOC exchanged during the maize growing season, only BVOC exchanges reported for other crops during their growth period were taken in account, i.e., harvest-induced emissions were not considered for comparison.

In our opinion, a comparison of BVOC budgets is the best way to evaluate the relative importance of a crop in terms of BVOC exchanges, because the duration of growing seasons differs greatly from one crop to another (and also under different

cultural conditions for the same crop). We reported in Table 5 the budget estimated for the maize field at LTO. The limited information on BVOC budgets that we found in the literature for other crops and grasses (also listed in Table 5), however, does not allow us to draw conclusions about the relative importance of maize in our region. The budgets reported by Crespo et al. (2013) and Graus et al. (2013) relate to different regions, in addition to which they were estimated from BVOC flux



measurements conducted over a few days and under a narrow range of weather conditions, making them highly uncertain. The budget reported by Bamberger et al. (2014) includes grass-cutting events, whereas harvest-induced BVOC emissions were not considered at LTO.

We therefore compared the BVOC flux ranges and averages obtained at LTO with those of other European croplands and
grasslands (Table 4). From the little information we found, we observed that for all investigated compounds the flux range observed at LTO was more than twice as low as that observed for *Miscanthus* (Copeland et al., 2012), another C4 crop. In addition, the methanol flux average measured at LTO was 9 times lower than that observed for a grassland (Bamberger et al., 2010). In contrast, the maximal methanol emission rate observed at the LTO site was twice as high as that reported for white clover (Custer and Schade, 2007). Maize field BVOC exchanges therefore seem to be smaller that those for other crops
grown in Europe, but this result should be further confirmed by other BVOC exchanges studies performed on other crops in our region.

### 3.6 BVOCs exchanged by a maize field under standard environmental conditions

Up-scaling models (Guenther et al., 2012 for MEGAN v2.1; Lathière et al., 2006 for ORCHIDEE) consider BVOC emissions from growing plants as a function of (i) an SEF that represents the mean emission of a particular plant functional
type under standard environmental conditions, and (ii) a multiplicative factor depending on PPFD, T and plant phenology, which reflects the response in emissions to varying environmental conditions.

The SEF values used in current up-scaling models for C4 crops do not match those observed at the LTO site for any model or any compound (Table 6). In particular, the SEF values used for isoprene and monoterpenes were several orders of magnitude lower at LTO than those used by models. This difference was even greater for isoprene because the M69 signal
was associated with isoprene for SEF computation at LTO, whereas it was probably not solely due to isoprene (Sect. 2.2). The SEF value measured at LTO for methanol was 2-4 times lower than that used by up-scaling model, while the SEF value measured for acetaldehyde was 0-11 times lower. The SEF values measured for acetone and acetic acid on our site were respectively 2 and 10 times lower than values used by MEGAN v2.1, and 2 and 5 times higher than values used by ORCHIDEE.
The SEF values used by up-scaling models rely on diverse BVOC flux measurement studies (see Guenther et al., 2012 and Lathière et al., 2006 for details). In particular, for Europe there is a comprehensive SEF inventory (Karl et al., 2009). Karl et al. (2009) noticed, however, that the SEFs given for croplands are default values because of the lack of information for those ecosystems. With regard to methanol, Stavrakou et al. (2011) used only one SEF derived from alfalfa for all croplands, although this species accounts for only 1% of the cultivated area worldwide (FAOSTATS). In contrast, maize is the second
most important crop and the most important C4 crop worldwide, accounting for 13 and 67% of the total cultivated area, respectively. In north-western Europe, it accounts for 12 and 99% of the total / C4 crop cultivated area, respectively. The C4 crop plant functional type considered by models can therefore be realistically equated with maize, especially in our region.





We would therefore advise modellers to use the SEFs reported in this study when estimating BVOC exchanges from C4 crops in north-western Europe.

## 4 Conclusions

This work constitutes the first BVOC study performed in a maize field over a whole growing season. It showed that the maize field emitted mainly methanol. Smaller but significant bi-directional OVOC exchanges were also recorded, resulting in a net emission of methanol and MEK and in a net uptake of acetic acid, acetaldehyde and MVK+MACR during the maize growing season. Terpenes exchange (mostly emissions) and a small but signifcant benzene and toluene uptake were also observed. Exchanges occurred throughout the growing season and each compound had different dynamics.

The observations at LTO showed in particular that: (i) the soil was an important BVOC source and sink; (ii) the BVOC exchanges were much lower than in other maize field studies, even when normalized by T, PPFD and phenology; (iii) they were also lower than those of other crops grown in Europe; and (iv) the estimated SEFs were much lower than those currently used by up-scaling models for the C4 crop plant functional type, of which maize is the main species.

Soil BVOC exchanges were as important, if not more so, as plant BVOC exchanges when the soil was bare and they decreased when maize was grown. The contribution of soil exchanges was probably particularly important on our site because BVOC exchanged by maize at LTO were small compared with those reported for other crops and grasses. Nevertheless, this work demonstrates that soil is a major actor in ecosystem-scale BVOC exchanges for some ecosystems. Future ecosystem-scale BVOC studies, particularly those investigating croplands, should therefore consider soil as a potential major BVOC reservoir. In addition, the BVOC exchange mechanisms between agricultural soils and the atmosphere need to be better understood in order to find out why these exchanges decrease during the maize growing season. Maize is cultivated in many regions of the world and is the main species of the C4 crop plant functional type. Our results showed, however, that the normalization of BVOC exchanges by T, PPFD and phenology was not enough to explain the huge difference in BVOC exchange rates among maize studies. This indicates that SEFs cannot be extrapolated to different world regions.

In this study, we proposed that our SEF values should be used for the C4 crop plant functional type. We also provided an estimation of the BVOC exchange budget of a maize field over a whole growing season. We argued that our values could be extrapolated to maize fields grown under similar agronomical and pedo-climatic conditions to those at LTO (i.e., in north-western Europe).

With the SEF values observed at LTO being far lower than those currently used by models, especially for terpenes, and with maize being the second most important cultivated crop, our results showed a reduced importance of BVOC emissions from croplands in our region. This indicates that deforestation and afforestation should result in even larger terpenes emission changes than currently estimated, especially in areas where maize production is important.





Specific maize SEF and BVOC exchange budget values should be obtained for other important agronomic regions by conducting long-term BVOC measurement studies similar to this one and by using the maize varieties and management practices commonly used in these regions. BVOC exchange mechanisms between maize fields and the atmosphere also need to be better understood in order to identify the reasons for the huge differences in normalized BVOC exchange rates

5    observed among maize studies and to discover if they behave according to up-scaling model algorithms beyond the standard conditions or to known OVOC exchange mechanisms (Niinemets et al., 2014). Given that each investigated compound had different exchange dynamics, mechanisms need to be evaluated separately for each compound, particularly for methanol on the LTO observations.

Finally, the BVOC exchange rates observed in this study were smaller than those observed in other crop studies in Europe,

10    suggesting that maize is a small BVOC exchanger crop in this region. Few BVOC measurement studies have been conducted under natural conditions in Europe on cropland ecosystems, however, and even less if we confine the comparison to north-western Europe. Future research should thus focus on other crops in order to extend the comparison. In particular, BVOC exchanges should be measured for winter wheat because, in terms of cultivated area, it is the most important crop in our region (FAOSTATS).

### Acknowledgements

The authors wish to thank Frédéric Wilmus, Henri Chopin and Alain Debacq for BVOC set-up installation and station monitoring, the farmer Mr. Van Eyck, the Lonzée-ICOS team for site follow-up and measurement of environmental parameters, and FNRS for its financial support (A2l5-MCF/DM-A362 FC 95918).

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



**Tables**

Table 1 m/z ratio of ion species measured at LTO and their potentially contributing compounds.

| m/z | Ion species | Potentially contributing compounds |
|-----|-------------|-----------------------------------|
| 21 | $H_3{}^{18}O^+$ | 3rd isotope of the 1st proton hydrate |
| 33 | $CH_5O^+$ | Methanol |
| 39 | $H_5{}^{16}O^{18}O^+$ | 3rd isotope of the 2nd proton hydrate |
| 45 | $C_2H_5O^+$ | Acetaldehyde (ACD) |
| 59 | $C_3H_7O^+$ | Acetone, propanal |
| 61 | $C_2H_5O_2{}^+$ | Acetic acid |
| 69 | $C_5H_9{}^+$ | Isoprene, methyl butenols, pentenols, methyl butanal |
| 71 | $C_4H_7O^+$ | Methyl vinyl ketone (MVK), methacrolein (MACR), pentanol |
| 73 | $C_4H_9O^+$ | Methyl ethyl ketone (MEK) |
| 79 | $C_6H_7{}^+$ | Benzene |
| 83 | $C_6H_{11}{}^+$ | Hexenols, hexenyl acetates, hexanal |
| 93 | $C_7H_9{}^+$ | Toluene |
| 137 | $C_{10}H_{17}{}^+$ | Monoterpenes (MT) |

Table 2 Maize phenological development. BBCH codification refers to Meier (2001). Both biomass and LAI refer to the beginning of a stage.

| | Stage | Date | | | BBCH | | | Biomass [gm$^{-2}$] | LAI [m$^2$m$^{-2}$] | Description of maize development |
|---|---|---|---|---|---|---|---|---|---|---|
| Vegetative growth | G (germination) | 14 May | - | 4 June | 00 | - | 14 | 0 | 0.00 | From sowing to 4-leaf stage Emergence on 25 May Considered as bare soil throughout the stage due to small biomass and biomass growth |
| | L (leaf unfolding) | 5 June | - | 26 June | 14 | - | 16[a] | 49 | 0.01 | From 4-leaf to 6-leaf stage Progressive canopy closure |
| | S (leaf area development and stem elongation) | 27 June | - | 5 August | 30 | - | 39 | 152 | 0.10 | From 6-leaf stage to panicle emergence Important increase in leaf area, maize height and biomass |
| Reproductive growth | R1 (inflorescence and fruit development) | 6 August | - | 29 August | 51 | - | 71 (73)[b] | 806 | 4.30 | From panicle emergence to fruit maturation Stamen appearance by 10 August (BBCH 61) Max. LAI reached by 20 August (5.06 m$^2$m$^{-2}$) Important biomass increase |
| | R2 (fruit maturation) | 29 August | - | 12 October | 71 (73)[b] | - | 89[c] | 1539 2019.2[d] | 5.06 | From fruit maturation to harvest Intermediate biomass increase |

[a]stages 17 to 19 did not occur

[b]Estimation: Fruit visible at the end of the stage, but exact grain maturity not identifiable

[c]stages 97 and 99 did not occur

[d]harvest





Table 3 BVOC composition per phenological stage. Each percentage corresponds to the ratio of the absolute mean flux of a particular compound during a phenological stage to the sum of the absolute mean fluxes of all investigated compounds during that stage.

|  | G | L | S | R1 | R2 | Whole season |
|---|---|---|---|---|---|---|
| Methanol | 55% | 56% | 76% | 66% | 31% | 66% |
| Acetaldehyde | 8% | 11% | 5% | 8% | 22% | 5% |
| Acetone | 7% | 6% | 0% | 2% | 5% | 0% |
| Acetic acid | 15% | 11% | 4% | 7% | 16% | 12% |
| M69 | 9% | 4% | 6% | 6% | 3% | 7% |
| MVK+MACR | 0% | 3% | 3% | 4% | 4% | 4% |
| MEK | 2% | 5% | 2% | 1% | 3% | 1% |
| Monoterpenes | 1% | 1% | 2% | 2% | 5% | 1% |
| Benzene | 1% | 3% | 0% | 1% | 4% | 2% |
| Toluene | 2% | 0% | 1% | 3% | 7% | 2% |



Table 4 OVOC exchanges by diverse crops and grass species.

| Flux [µgm⁻²h⁻¹] | C4 crops | | | Grasses | | | | | Soil |
| --- | --- | --- | --- | --- | --- | --- | --- | --- | --- |
| Ecosystem | Maize | | | Miscanthus | | White clover | Mixed grasses | | Cambisoil |
| Species | Avg ± SE | Min-Max | | | | | | | |
| Methanol | 27 ± 0.8 | -342 - 708 | 3450 ± 1456[a] | 821 ± 59[b] | -2000 - 3000 | -213 - 320 | 131 - 1073 | 288[c] | 0 - 533 |
| Acetaldehyde | -2 ± 0.3 | -179 - 155 | | 159 ± 54[b] | -1000 - 1000 | | | | |
| Acetone | -0.1 ± 0.2 | -198 - 265 | 425 ± 223[a] | 125 ± 10[b] | -2000 - 2000 | | | | -48 - 242 |
| Acetic acid | -5 ± 0.2 | -347 - 206 | | 380 ± 57[b] | -1000 - 500 | | -41 - -35 | | |
| Author(s) | This study | | Das et al. (2003) | Graus et al. (2013) | Copeland et al. (2012) | Custer and Schade (2007) | Ruuskanen et al. (2011) | Bamberger et al. (2010) | Schade and Custer (2004) |
| Location | Belgium | | North Carolina | Colorado (USA) | United Kingdom | Germany | Austria | Austria | Germany |
| Meas. scale | Ecosystem | | Ecosystem | Leaf | Ecosystem | Ecosystem | Ecosystem | Ecosystem | Ecosystem |
| Meas. technique | DEC-MS | | Gradient | In situ cuvette | DEC-MS | DEC-MS | EC | DEC-MS | DEC-MS |
| BVOC conc. Instrument | PTR-MS | | GC-MS | PTR-MS | PTR-MS | PTR-MS | PTR-MS-TOF | PTR-MS | PTR-MS |
| Meas. Duration [days] | 148 | | 4 | 2 | 30 | 40 | 2 | 155 | 5 |
| Temperature range [°C] | 4-29 | | 24-27 | 30[c] | 1-29 | 15-30 | 1-29 | 16[c] | 20-40 |

[a] midday fluxes
[b] fluxes up-scaled with an LAI of 6m²m⁻² (Graus et al., 2013)
[c] avg. growing season only





Table 5 BVOC budget estimated for maize at LTO and on diverse crops over a whole growing season.

| [$g_{BVOC}ha^{-1}$] Crop | Methanol | Acetaldehyde | Acetone | Acetic acid | Author |
|---|---|---|---|---|---|
| Maize | $960 \pm 29$ | $-70 \pm 9$ | $-1 \pm 8$ | $-181 \pm 8$ | This study |
| | 5521[a] | 1075[a] | 838[a] | 2251[a] | Graus et al. (2013) |
| Elephant grass | 20000 | 30000 | 37000 | | Crespo et al. (2013) |
| Miscanthus | 3780[b] | 680[b] | 1180[b] | 3580[b] | Graus et al. (2013) |
| Grassland | 22171[c] | 101[c] | 200[c] | | Bamberger et al. (2014) |

[a] Original data in $g_{BVOC}L_{ethanol}^{-1}$. Converted with an ethanol yield of 0.38 $L_{ethanol}kg_{grain}^{-1}$, a grain:residue ratio equal to 1:1, and a biomass yield of 9.59Mgha$^{-1}$, as used by Graus et al. (2013).

[b] Original data in $g_{BVOC}L_{ethanol}^{-1}$. Converted with an ethanol yield of 0.4 $L_{ethanol}kg_{leaves}^{-1}$ and a leaf biomass of 5Mgha$^{-1}$, as used by Graus et al. (2013).

[c] Annual budget. Average of 2009 and 2011 budgets for methanol, 2009 budget only for other compounds. Original data in mg C m$^{-2}$yr$^{-1}$, converted with the molar mass of each individual compound.

Notice that budgets reported for grassland include both growing periods and cutting events.



Table 6 SEF recorded at LTO under standard environmental conditions and used in up-scaling models for C4 agricultural crops.

| | This study | | MEGAN v2.1[d] | ORCHIDEE[c] |
|---|---|---|---|---|
| | $[\mu g\ m_{soil}^{-2}\ h^{-1}]^a$ | $[\mu g\ g_{dw}^{-1}\ h^{-1}]^b$ | $[\mu g\ m_{soil}^{-2}\ h^{-1}]$ | $[\mu g\ g_{dw}^{-1}\ h^{-1}]$ |
| Isoprene | 8[f] ± 5 | 0.058[f] ± 0.038 | 200 | 8.500 |
| Monoterpenes | 4 ± 6 | 0.030 ± 0.040 | 2 | 0.227 |
| Methanol | 231 ± 19 | 1.642 ± 0.137 | 500[d]/800[e] | 2.667 |
| Acetone | 46 ± 8 | 0.324 ± 0.057 | 80 | 0.113 |
| Acetaldehyde | 7 ± 9 | 0.046 ± 0.065 | 80 | 0.046 |
| Acetic acid | 8 ± 9 | 0.055 ± 0.072 | 80 | 0.013 |

[a] Standard environmental conditions chosen to match MEGAN v2.1 standard environmental conditions (Guenther et al., 2012): 28°C ≤ T° ≤ 32°C; PPFD ≥ 1000 $\mu molm^{-2}s^{-1}$; 4.38 $m^2m^{-2}$ ≤ LAI ≤ 5.04 $m^2m^{-2}$.

[b] Specific leaf weight used for the conversion from $m_{soil}^{-2}$ to $g_{dw}^{-1}$: 29.0 $g_{DW,leaf}\ m_{leaf}^{-2}$ (meas. performed on a mature maize leaf of variety Prosil, Mozzaffar et al., 2015, pers. comm.). LAI used for conversion : 4.85 $m^2m^{-2}$

[c] Lathière et al. (2006.). SEF are here given in $\mu g_{Compound}$ instead of $\mu g_C$

[d] Guenther et al. (2012)

[e] Stavrakou et al. (2011)

[f] Derived from M69 flux





**Figures**

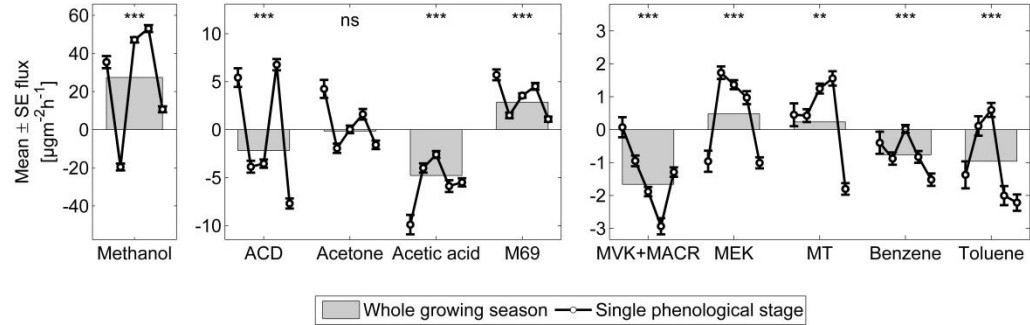

Figure 1 BVOC exchanges over the growing season. Phenological stages, from left to right: G, L, S, R1 and R2. Stars indicate the significance level of flux averages over the whole growing season. See Table 1 for compound abbreviations and
5   Table 2 for phenological stage description. Note the varying scales used for the y axes.





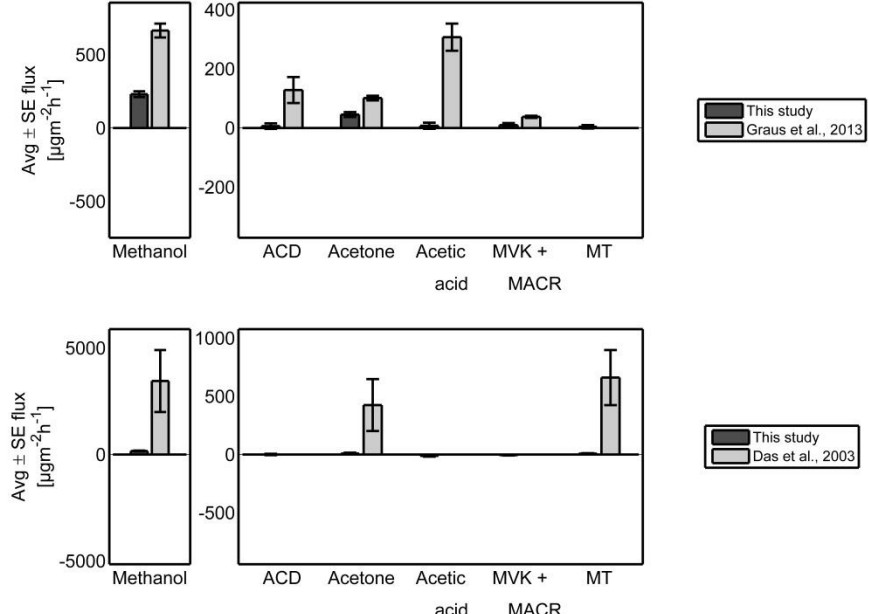

Figure 2 Comparison of BVOC exchanges at LTO with other maize studies. For comparison with Graus et al. (2013): R1 stage, $28°C \leq T° \leq 32°C$; PPFD $\geq 1000$ μmolm$^{-2}$s$^{-1}$. BVOC exchanges reported by Graus et al. (2013) were up-scaled with LAI measured at LTO during the R1 stage (4.86 m$^2$m$^{-2}$). For comparison with Das et al. (2003): L and S stages, $24°C \leq T° \leq 28°C$, fluxes taken from 10:30 to 13:30 UTC in order to capture midday fluxes. See Table 1 for compound abbreviations and Table 4 for maize studies description. Note the varying scales for the y axes.