# Peer review of "Are BVOC exchanges in agricultural ecosystems overestimated? Insights from fluxes measured in a maize field over a whole growing season"

_Atmospheric Chemistry and Physics, 2015_

## Referee Comment (RC1) · Anonymous Referee #1 · 21 Feb 2016

General comments: The focus of this paper is to investigate BVOC exchange on a maize field via comprehensive in situ measurements so as to examine previous results and BVOC emission models. The major conclusions from the authors were that BVOC exchange fluxes in the maize field was lower considerably than those measured in other crops. As a result, a BVOC emission model created from standard emission factors seemed to overestimate BVOC emission fluxes and hence such the model should treat BVOC emissions case by case in different crops field. The authors further recommended to incorporate their SEF obtained from this field study in BVOC emission modeling. The evidence from their field study was strong and their arguments in the presentation were also reasonable. I recommend acceptance for publication in ACP

after clarifying following questions.

Specific Comments: Pg. 3. section 2.1.2 Flux measurements; pg. 5, section 2.3 BVOC flux computation

I would suggest authors to give the expression of BVOC flux equation which should be the product of measured concentration and 'vertical velocity'. I would assume that '3D sonic anemometer' measures turbulent fluctuations of vertical wind, not vertical wind itself?

Pg. 11. 'Given the huge differences in normalized BVOC exchange rates among studies, we conclude... by normalizing T and PPFD'. Can BVOC exchange rate be normalized by solar zenith?

Pg. 6, ' according to a lower u* threshold'. What is 'lower u* threshold'?

Pg. 10, 'the methanol and acetaldehyde fluxes measured at our site were of the same order of magnitude for bare soil as for fully developed vegetation '; pg. 11, 'the soil was an important BVOC source and sink'. What is net flux of BVOC over bare soil?

Technical corrections: Pg. 1, line 17, 'developped" is 'devloped' Pg. 1, line 22, 'in this in this', delete one 'in this' Pg. 8, line 15, 'where' is 'when'?

---

## Referee Comment (RC2) · Anonymous Referee #2 · 26 Feb 2016

**1   General comments**

This study reports new measurements of BVOC fluxes over a maize field in Belgium using eddy covariance. The authors are the first (to their knowledge) to observe a full growing season, and also the first to measure at a European site, making this data set a valuable addition to the extremely limited database of BVOC flux observations in maize fields.  Compared to past cropland studies, including two American-based maize studies, they observed similar BVOC composition – methanol (dominant), acetic acid, acetone, acetaldehyde, terpenes – but concentrations were significantly lower in the present study.  The emission factors prescribed for crops, as used in BVOC

emission models, were higher than those estimated from the field measurements in this study. From these results, the authors conclude that BVOC exchanges from maize vary regionally around the world, and that emission factors in models should account for this variability. The emission factors estimated in this study are recommended as representative of C4 crops in north-western Europe. With data for the full growing season, the authors were able to quantify the relative contributions of soil and plant to BVOC fluxes, finding that the soil (bare soil in particular) contributed about as much as vegetation.

This study, appropriate for ACP, makes a substantial contribution to the sorely limited observational record of BVOC fluxes in the ever-growing maize landscape. The purpose and goal is well articulated with strong motivating support. The methods are complete and clearly described. The conclusions drawn from the results follow logically, though may be overstated given still limited data and large uncertainties. Overall, the manuscript is well-written and merits publication in ACP provided the following comments are considered.

**2 Specific comments**

- Pg 2, Lns 13-17: What is the relative contribution from crops relative to other biogenic VOC sources (e.g., forests)? Any estimates on maize specifically? Though corn covers a large landscape, are emission rates large enough to significantly contribute to the global VOC budget?

- Pg 2, Lns 13-19: What did Das et al. and Graus et al. find? (i.e. the baseline knowledge going into this study)

- Pg 2, Ln 25: define "standard"

- Pg 3, Ln 26: Replace "a few" with a numeric range, if possible.

- Pg 7, Ln 3: Combine with previous paragraph.

- Pg 8, Ln 26 and 28: What are "normal" (line 26) and "natural" (line 28) weather conditions for this area?

- Pg 9, Ln 22: Do plants take up monoterpenes? Are they not primarily emitted? Perhaps downward flux doesn't necessarily signify uptake in this case?

- Pg 9, Ln 23-26: Consider revising these paragraph breaks.

- Pg 9, Ln 27 - Pg 10, Ln 4: These lines seem to belong in Sec 3.5.1.

- Pg 11, Ln 1-2: The maize field area seems relatively small (155 x 255 m, Pg 3, Ln 4) and makes me wonder about the possibility of advection bringing in low-VOC air, thus resulting in lower VOC than other studies. Are the field sites of Das et al. and Graus et al. much larger and thus less influenced by outside air? Any correlations with wind direction that suggest advective influences? How does the flux footprint at the measurement heights on Pg 3, Lns 17-18 compare with size of the field (185 x 255 m) on Pg 3, Ln 4? Of course, if there is an advective signal, this puts into the question the validity of the horizontal homogeneity assumption and challenges whether this is representative of north-western Europe or "ecosystem-scale."

- Pg 12, Ln 27: What are the "default values" and what are they based on?

- Pg 13, Ln 1-2: Are you comfortable advising modelers to use these SEFs given all the uncertainties? Given the limited data, and the large discrepancies with the two studies cited here, I feel more data is needed to validate the SEFs found here before they are deemed a reliable representation of C4 crops in NW Europe. Instead, you might advise modelers to be wary that current SEFs may be overestimates and advise them to include that potential caveat in their studies.

- Pg 13, Ln 24: Again, are you confident enough in your SEFs to say they "should" be used to represent C4 crop PFT?

- Sec 4 (Conclusions): Can you draw any new conclusions about the evolution of BVOC fluxes from maize fields throughout the growing season now that you have this new data set that didn't exist before? For instance, can you comment on the variability throughout the season in Figure 1 and how the "plant phenology" dependence of modeled emissions (Pg 12, Ln 15) captures that variability?

**3  Technical corrections**

- Pg 1, Ln 16: "as" –> "on the"?

- Pg 1, Ln 17: "developped" –> "developed"

- Pg 1, Ln 22: extra "in this"

- Pg 1, Ln 28: "affect" –> "effect"

- Pg 2, Ln 1: add "the" before "formation"

- Pg 2, Ln 11: add "a" before "few"

- Pg 2, Ln 15: spell out FAOSTATS

- Pg 4, Ln 9: "There is more ... (2009)." –> "See Aubinet et al. (2009) for more ... set-up."

- Pg 7, Ln 25: "2h" –> "2 h"

- Pg 8, Ln 10: spell out BBCH, and define

- Pg 10, Ln 11: "reporded" –> "reported"

- Pg 12, Ln 21: "model" –> "models"

- Pg 13, Ln 7: "signifcant" –> "significant"

---

## Author Comment (AC1) · 25 Mar 2016

**Author's reply to anonymous referee #1**

*Reply for the revision process of the manuscript untitled "Are BVOC exchanges in agricultural ecosystems overestimated? Insights from fluxes measured in a maize field over a whole growing season" published on ACPD.*

First of all, we would like to thank the referee for her/his comments and suggestions which contribute to improve the quality of this manuscript. We answered all general and specific comments point-by-point as thoroughly as possible and adapted the manuscript accordingly.

Technical comments consisted of spelling mistakes and minor phrase structure, so they did not request a detailed author's reply. Consequently, we did not include them in this document. They will of course be taken into full consideration during the manuscript revision, and the text will be corrected accordingly.

The reply document is formatted as follows:

Comment n°X where X is a number is the comment number;

PX LY corresponds to the line Y in the page X; Sec. X corresponds to the section X;

**Referee comment;**

Author's reply;

*Author's changes in the manuscript.* Original text and revised text are detailed.

Page and line indexes after each comment number, in the author's reply and after the mention "original text" refer to the discussion manuscript published in ACPD.

Page and line indexes after the mention "revised text" refer to the revised manuscript to be sent to the editor.

References used to answer the referee comments were listed in "Author's reply references". As all these references were present in the discussion paper, no additional reference will be included in the revised paper following the referee #1 comments.

**General comment**

**The focus of this paper is to investigate BVOC exchange on a maize field via comprehensive in situ measurements so as to examine previous results and BVOC emission models. The major conclusions from the authors were that BVOC exchange fluxes in the maize field was lower considerably than those measured in other crops. As a result, a BVOC emission model created from standard emission factors seemed to overestimate BVOC emission fluxes and hence such the model should treat BVOC emissions case by case in different crops field. The authors further recommended to incorporate their SEF obtained from this field study in BVOC emission modeling. The evidence from their field study was strong and their arguments in the presentation were also reasonable. I recommend acceptance for publication in ACP after clarifying following questions.**

The authors are very grateful to the referee for her/his positive comment. We adapted the manuscript following her/his questions and suggestions to make it clearer.

**Specific comments**

Comment n°1 (P3 Sec. 2.1.2; P5 Sec. 2.3)

**I would suggest authors to give the expression of BVOC flux equation which should be the product of measured concentration and 'vertical velocity'. I would assume that '3D sonic anemometer' measures turbulent fluctuations of vertical wind, not vertical wind itself**?

The 3D sonic anemometer measures the wind velocity at high frequency in 3 non parallel directions. So it gives measurements of the vertical wind speed component at high frequency; those measurements are also called instantaneous vertical wind speed w. w is then separated into two terms:

$$w = \bar{w} + w'$$

Where $\bar{w}$ corresponds to the mean vertical component of the wind speed, computed by averaging w over each half-hour, and w' corresponds to the fluctuations of the vertical wind speed component around this mean.

As this is well-known information, we will not detail this expression. However, to make the text clearer, we will explicitly write that the BVOC fluxes were calculated from the covariance between the BVOC concentration and the vertical component of the wind speed, both being measured at high frequency.

Original text (P3 L13-14): *The BVOC fluxes were computed every half-hour from high frequency vertical wind speed and BVOC concentration measurements using the disjunct eddy covariance by mass scanning (DEC-MS) technique.*

Revised text (P3 L14-15): *The BVOC fluxes were computed every half-hour using the disjunct eddy covariance by mass scanning (DEC-MS) technique, i.e. from the covariance between the vertical component of the wind speed and the BVOC mixing ratio, both variables being measured at high frequency.*

Comment n°2 (P11 L16)

**Given the huge differences in normalized BVOC exchange rates among studies, we conclude [...] by normalizing T and PPFD'. Can BVOC exchange rate be normalized by solar zenith?**

We used PPFD for normalisation as this is done by other authors measuring in the field (e.g. Park et al., 2014). But indeed, standard conditions defined by the up-scaling models are rather defined for particular solar zenith angle and PPFD transmission ratio (Guenther et al., 2006). We prefer to keep PPFD as the normalizing factor because we think that use of solar angle may bring uncertainties. Indeed, in Graus et al., 2013 and Das et al., 2003 articles, we did not find information about the solar angle and the PPFD transmission ratio. And we did not find enough information in those articles to estimate these values with accuracy. Consequently, we prefer keeping the normalisation by PPFD, since it is based on data given by the authors themselves, in order to rely on known values when comparing data.

Comment n°3 (P6 L9)

**'According to a lower $u_*$ threshold'. What is 'lower $u_*$ threshold'?**

The friction velocity, represented by the symbol "$u_*$", provides insights about the importance of turbulent processes on the site. Flux data measured by the eddy covariance technique are only valid when tracers are carried from the atmosphere to the ecosystem through turbulent exchange processes. Consequently, flux data are not representative anymore of the actual exchange between the ecosystem and the atmosphere when the turbulence is not important enough. Practically, we use $u_*$ measurements to determine whether the turbulence is sufficient so that fluxes measured by the eddy covariance technique are valid. The value of $u_*$ above which the turbulence is sufficient is then called the 'lower $u_*$ threshold'.

Following your question, we will clarify in the section about friction velocity that flux data which were measured at $u_*$ values below a certain threshold must theorically be discarded for non-soluble compounds.

Original text (P6 L9-10): *It should be noted that we did not filter BVOC fluxes according to a lower $u_*$ threshold or to stationarity. Indeed, $u_*$ can actually control soluble BVOC fluxes (*Aubinet et al., 2012*;Laffineur et al., 2012).*

Revised text (P6 L10-13): *It should be noted that we did not filter BVOC fluxes below a certain $u_*$ threshold or according to stationarity. Theoretically, for non-soluble compounds, when measuring fluxes by the EC technique, flux data which were measured at $u_*$ values below a certain threshold must be discarded (Aubinet et al., 2012). However, we did not apply this specific filtering criterion because $u_*$ can actually control soluble BVOC fluxes (Aubinet et al., 2012;Laffineur et al., 2012).*

Comment n°4 (P10 L14 and P11)

**'The methanol and acetaldehyde fluxes measured at our site were of the same order of magnitude for bare soil as for fully developed vegetation '; 'the soil was an important BVOC source and sink'. What is net flux of BVOC over bare soil?**

The second sentence is not present in P11. We guess you refered to P13 L9? Using the eddy covariance technique, what we actually measure is the net flux between the ecosystem and the atmosphere. But we observed that for some compounds like methanol, most net fluxes were positive when the soil was bare. This means that for most data, there were net methanol emissions from the ecosystem to the atmosphere. From this we concluded that there were methanol sources in the ecosystem. On the opposite, for other compounds, such as acetic acid, most net fluxes were negative when the soil was bare, meaning that for most data there were net acetic acid uptakes from the atmosphere to the ecosystem. Then we concluded that there were acetic acid sinks in the ecosystem.

When we wrote 'the soil was an important BVOC source and sink', we intended to indicate that the soil was a source for some BVOC compounds while it was a sink for other BVOC compounds. The use of the terms "source" and "sink" without mentioning that they refered to different compounds was however probably confusing. In order to avoid any further confusion, we will complete this sentence.

Orignal text (P13 L9): *the soil was an important BVOC source and sink*.

Revised text (P14 L25-26): *the soil was an important methanol and acetaldehyde source, and an important acetic acid sink*.

**Author's reply references**

Aubinet, M., Vesala, T. and Papale, D.: Eddy covariance a practical guide to measurement and data analysis, Springer, Dordrecht; New York., 2012.

Das, M., Kang, D., Aneja, V. P., Lonneman, W., Cook, D. R. and Wesely, M. L.: Measurements of hydrocarbon air-surface exchange rates over maize, Atmos. Environ., 37(16), 2269–2277, 2003.

Graus, M., Eller, A. S. D., Fall, R., Yuan, B., Qian, Y., Westra, P., de Gouw, J. and Warneke, C.: Biosphere-atmosphere exchange of volatile organic compounds over C4 biofuel crops, Atmos. Environ., 66, 161–168, 2013.

Guenther, A., Karl, T., Harley, P., Wiedinmyer, C., Palmer, P. I. and Geron, C.: Estimates of global terrestrial isoprene emissions using MEGAN (Model of Emissions of Gases and Aerosols from Nature), Atmos. Chem. Phys., 6(11), 3181–3210, doi:10.1029/2003GL017336; Adams, J., Constable, J., Guenther, A., Zimmerman, P., An estimate of natural volatile organic compound emissions from vegetation since the last glacial maximum (2001) Chemosphere- Global Change Science, 3, pp. 73-91; (1998) Global Change Scenarios of the 21st Century. Results from the IMAGE 2.1 Model, p. 296. , Alcamo, J., Leemans R., and Kreileman, E. (Eds.):

Pergamon & Elseviers Science, London; Alessio, G.A., De Lillis, M., Fanelli, M., Pinelli, P., Loreto, F., Direct a, 2006.

Laffineur, Q., Aubinet, M., Schoon, N., Amelynck, C., Müller, J. F., Dewulf, J., Van Langenhove, H., Steppe, K. and Heinesch, B.: Abiotic and biotic control of methanol exchanges in a temperate mixed forest, Atmos. Chem. Phys., 12(1), 577–590, doi:10.1016/j.ijms.2004.08.012, 2012.

Park, J. H., Fares, S., Weber, R. and Goldstein, A. H.: Biogenic volatile organic compound emissions during BEARPEX 2009 measured by eddy covariance and flux-gradient similarity methods, Atmos. Chem. Phys., 14(1), 231–244, doi:10.5194/acp-9-5505-2009, 2014.

---

## Author Comment (AC2) · 25 Mar 2016

**Author's reply to anonymous referee #2**

*Reply for the revision process of the manuscript untitled "Are BVOC exchanges in agricultural ecosystems overestimated? Insights from fluxes measured in a maize field over a whole growing season" published on ACPD.*

First of all, we would like to thank the referee for her/his comments and suggestions which contribute to improve the quality of this manuscript. We answered all general and specific comments point-by-point as thoroughly as possible and adapted the manuscript accordingly.

Technical comments which consisted of spelling mistakes or minor phrase structure were not included in this document, as they did not request a detailed author's reply. They will of course be taken into full consideration during the manuscript revision, and the text will be corrected accordingly. Technical comments which requested some author's reply were discussed after specific comments.

The reply document is formatted as follows:

Comment n°X where X is a number is the comment number;

PX LY corresponds to the line Y in the page X; Sec. X corresponds to the section X;

**Referee comment;**

Author's reply;

*Author's changes in the manuscript.* Original text and revised text are detailed.

Page and line indexes after each comment number, in the author's reply and after the mention "original text" refer to the discussion manuscript published in ACPD.

Page and line indexes after the mention "revised text" refer to the revised manuscript to be sent to the editor.

References used to answer the referee comments were listed in "Author's reply references". As these references were present in the discussion paper or were not added in the revised texts, no additional reference will be included in the revised paper following the referee #2 comments.

**General comment**

**This study reports new measurements of BVOC fluxes over a maize field in Belgium using eddy covariance. The authors are the first (to their knowledge) to observe a full growing season, and also the first to measure at a European site, making this data set a valuable addition to the extremely limited database of BVOC flux observations in maize fields. Compared to past cropland studies, including two American-based maize studies, they observed similar BVOC composition – methanol (dominant), acetic acid, acetone, acetaldehyde, terpenes – but concentrations were significantly lower in the present study. The emission factors prescribed for crops, as used in BVOC emission models, were higher than those estimated from the field measurements in this study. From these results, the authors conclude that BVOC exchanges from maize vary regionally around the world, and that emission factors in models should account for this variability. The emission factors estimated in this study are recommended as representative of C4 crops in north-western Europe. With data for the full growing season, the authors were able to quantify the relative contributions of soil and plant to BVOC fluxes, finding that the soil (bare soil in particular) contributed about as much as vegetation. This study, appropriate for ACP, makes a substantial contribution to the sorely limited observational record of BVOC fluxes in the ever-growing maize landscape. The purpose and goal is well articulated with strong motivating support. The methods are complete and clearly described. The conclusions drawn from the results follow logically, though may be overstated given still limited data and large uncertainties. Overall, the manuscript is well-written and merits publication in ACP provided the following comments are considered.**

The authors are very grateful to the referee for her/his positive comment. We adapted the manuscript following her/his questions and suggestions to make it clearer, and we moderated some conclusions following her/his comments.

**Specific comments**

Comment n°1 (P2 L13 to P2 L 17)

**What is the relative contribution from crops relative to other biogenic VOC sources (e.g., forests)? Any estimates on maize specifically? Though corn covers a large landscape, are emission rates large enough to significantly contribute to the global VOC budget?**

Crops are considered by up-scaling models (Guenther et al., 2012) as smaller BVOC emitters than forests for terpenes, but as equal OVOC emitters.

At our knowledge, the only BVOC exchanges rates estimates available for maize come from Graus et al., 2013, Das et al., 2003 and this study. Das et al., 2003 concluded that maize is a major BVOC source and could significantly contribute to the air quality in regions where it is widely cropped, whereas Graus et al., 2013 concluded that maize contribution to air quality was small. On our site, we observed that maize was a small BVOC exchanger in comparison with other crops and grasses, thereby suggesting a negligible contribution of maize to air quality through BVOC exchanges.

However, to our opinion, evaluating the actual corn influence on the global BVOC budget remains currently very uncertain. First, there have been few studies dedicated to that crop and the observed BVOC exchanges rates strongly differed among studies. Second, there are strong differences in maize phenology and in maize growing season length among world regions. Consequently, the BVOC maize budget over its growing season may vary strongly among world regions. Therefore, before being able to answer the question, more long-term measurements studies focusing on maize should be conducted in different parts of the world to estimate the BVOC budget estimation in each region.

Comment n°2 (P2 L13 to P2 L19)

**What did Das et al. and Graus et al. find? (i.e., the baseline knowledge going into this study)**

Das et al., 2003 found high methanol and acetone emissions from maize and suggested that maize could play a great role in atmospheric chemistry in regions where maize is abundantly cropped, like the Corn Belt in USA.

Graus et al., 2013 identified some compounds emitted or taken up by the maize, and estimated the maize emission rate per liter of produced bio-ethanol. They concluded that maize did not play a great role in the atmospheric chemistry. The BVOC exchange rates they observed were lower than those observed by Das et al., 2003; they suggested a potential leaf age effect on emissions to explain those differences.

We did not mention all these findings in the "Introduction" section of the manuscript because we discussed them in the "Result and discussion" parts. Following your question, though, we will introduce them in the "Introduction" section in order to provide insights about the "baseline knowledge" to the reader.

Original text (P2 L13 to P2 L19): *So far as we know, only two BVOC measurement studies have dealt with maize (Das et al., 2003; Graus et al., 2013) [...] In addition, both maize studies were conducted over only a few days and under poorly contrasted weather conditions, and were thus unable to evaluate the relative effects of climate and phenology on BVOC exchanges. Knowledge about BVOC exchanges from maize therefore remains very limited.*

Revised text (P2 L13 to P2 L24): *So far as we know, only two BVOC measurement studies have dealt with maize (Das et al., 2003; Graus et al., 2013) [...] Graus et al., 2013 determined the BVOC exchange composition of maize leaves, and estimated the maize BVOC budget by up-scaling and extrapolating the BVOC fluxes they*

*measured to the whole growing season. Das et al., 2003 found large methanol and acetone emissions from maize and suggested that that crop could play an important role in the atmospheric chemistry in regions where it is widely present, e.g. the Corn Belt zone in USA.*

*However, both studies were conducted over only a few days and under poorly contrasted weather conditions. Consequently, they were unable to evaluate the relative effects of climate and phenology on BVOC exchanges, so that the current estimated maize BVOC budget is uncertain. Knowledge about BVOC exchanges from maize therefore remains very limited.*

Comment n°3 (P2 L25)

**Define "standard"**

Standard conditions refer to the standard conditions defined by up-scaling models. More particularly, we relied on the standard conditions described in Guenther et al., 2006. This will be explicitely mentioned in the revised manuscript.

Original text (P2 L25): *What quantity of BVOCs is exchanged in a maize field under standard environmental conditions?*

Revised text (P2 L30-32): *What quantity of BVOCs is exchanged in a maize field under the standard environmental conditions (standard conditions correspond to the environmental conditions defined for the MEGAN up-scaling model in Guenther et al., 2006)?*

Comment n°4 (P3 L26)

**Replace "a few" with a numeric range, if possible.**

This will be done in the revised manuscript version.

Original text (P3 L26): *In order to prevent water vapour condensation in the main sampling line, [...], the sampling line was thermally insulated and heated a few degrees above ambient temperature.*

Revised text (P3 L28): *In order to prevent water vapour condensation in the main sampling line, [...], the sampling line was thermally insulated and heated on average 2.6°C above the ambient temperature.*

Comment n°5  (P7 L3)

**Combine with previous paragraph.**

This will be done in the revised manuscript version.

Comment n°6 (P8 L26 and P8 L28)

**What are "normal" (L26) and "natural" (L28) weather conditions for this area?**

 Normal conditions are defined by the Royal Meteorological Institute of Belgium. They correspond to averaged conditions observed at the meteorological station of Uccle (Belgium) over a period of 30 years (1981-2010). The "normal" mean temperature in Belgium is 3.6°C, 10.1°C, 17.5°C and 10.9°C while the "normal" precipitation is 220.5 mm, 187.8 mm, 224.4 mm and 219.9 mm for winter, spring, summer and fall, respectively (*Résumé climatologique de l'année 2012* published on the RMI website: http://www.meteo.be/meteo/view/fr/10275209-2012.html). These conditions will be mentioned in the revised manuscript.

By the term "natural conditions", we mean environmental conditions that occur under real field conditions, in contrast to lab conditions or cuvette conditions. We will replace this term by "real" to be less confusing.

Note that, following the comments n°12 and 13, the text in the Sec. 3.2 was modified and the section was moved after the Sec 3.5.

Original text (P8 L26): *The weather conditions during the study were among normal for the time and place (Royal Meteorological Institute, Belgium).*

Revised text (P14 L5-9): *The weather conditions during the study were normal for the time and place (normal conditions are defined by the Belgian Royal Meteorological Institute by averaging records taken in Uccle, Belgium over the period 1981-2010. It corresponds to 3.6°C, 10.1°C, 17.5°C and 10.9°C and to 220.5 mm, 187.8 mm, 224.4 mm and 219.9 mm cumulated precipitation in winter, spring, summer and fall, respectively).*

Original text (P8 L28): *The maize field grown at LTO was thus well representative of the fields of maize grown under natural weather conditions [...]*

Revised text (P14 L10): *The maize field grown at LTO was thus well representative of the fields of maize grown under real conditions [...]*

Comment n°7 P9 L22

**Do plants take up monoterpenes? Are they not primarily emitted? Perhaps downward flux doesn't necessarily signify uptake in this case?**

Plants are mostly known to emit monoterpenes, but it has been demonstrated that some nonemitting plant species were also able to take up terpenes (Noe et al., 2008). Furthermore, significant monoterpenes deposition has also been observed at ecosystem-scale on mountain grassland (Bamberger et al., 2011). As a result, the occurrence of monoterpenes uptakes on our site is not something aberrant.

Then, we did not evaluate whether maize actually consumed these monoterpenes, or if there were other monoterpenes sinks in our ecosystem. This is the reason why we mentioned "monoterpenes uptakes" but never told about "plant monoterpenes uptakes" in the discussion manuscript.

However, in order to avoid any further confusion, we will specify that the uptakes were observed between the atmosphere and the field and not between the atmosphere and the maize itself.

Orginal text (P9 L22): *Significant emissions were found for both compounds during all stages, apart from R2, during which uptake of monoterpenes was observed instead.*

Revised text (P10 L1-2): *Significant emissions were found for both compounds during all stages, apart from R2, during which the monoterpenes have been taken up by the maize field ecosystem.*

Comment n°8 (P9 L23 to P9 L26)

**Consider revising these paragraph breaks.**

We suggest moving the L25 to L26 at the end of the Sec. 3.3.

Orginal text (P9 L25 to P9 L26): *Small but significant benzene and toluene uptake was observed for all phenological stages. Each compound contributed up to7% to the total BVOC exchange.*

*Besides, each investigated BVOC showed different seasonal dynamics, indicating that the sources, sinks strength and/or exchange mechanisms differed for each compound.*

*The BVOC exchange composition observed at LTO matched those observed on diverse croplands and grasslands fairly well [...] In addition, our observations disagree with the hypothesis proposed by White et al. (2009) that maize could be an important toluene source.*

Revised text (P10 L3 to P10 L18): *Small but significant benzene and toluene uptake was observed for all phenological stages. Each compound contributed up to 7% to the total BVOC exchange.*

*The BVOC exchange composition observed at LTO matched those observed on diverse croplands and grasslands fairly well [...] In addition, our observations disagree with the hypothesis proposed by White et al. (2009) that maize could be an important toluene source.*

*Lastly, each investigated BVOC showed different seasonal dynamics. This indicates that the sources, sinks strength and/or exchange mechanisms differed for each compound.*

Comment n°9 (P9 L27 to P10 L4)

**These lines seem to belong in Sec 3.5.1.**

The lines P9 L27 to P10 L4 intend to compare the BVOC composition between studies in a qualitative way (this is: what was emitted? What was taken up? Which were the major compounds? Did the maize field exchange the same compounds as other grassland and cropland...).

In contrast, the Sec 3.5.1. intends to compare the BVOC between studies in a quantitative way (this is: is the BVOC exchange rate similar to other studies ?).

We prefer keeping both discussions separated because they lead to different conclusions: qualitatively, the BVOC composition found at LTO is similar to other studies, but quantitatively, the exchanges rates were lower.

However, to clarify this distinction, we will indicate the terms "quantitative" and "qualitative" in the text or section title when relevant.

Original text (P9 L1): *BVOCs exchanged at LTO*

Revised text (P9 L11): *BVOCs exchange composition at LTO and qualitative comparison with other crops*

Original text (P9 L27): *The BVOC exchange composition observed at LTO matched those observed on diverse croplands and grasslands fairly well.*

Revised text (P10 L7): *Qualitatively, the BVOC exchange composition observed at LTO matched those observed on diverse croplands and grasslands fairly well.*

Original text (P10 L27): *Comparison of BVOCs exchanged at LTO with other maize BVOC studies*

Revised text (P11 L10): *Quantitative comparison of BVOCs exchange rates at LTO with other maize BVOC studies*

Original text (P11 L22): *Comparison of BVOCs exchanged at LTO with other crops*

Revised text (P12 L3): *Quantitative comparison of BVOCs exchange rates at LTO with other crops*

Comment n°10 (P11 L1 to P11 L2)

**The maize field area seems relatively small (155 x 255 m, Pg 3, L4) and makes me wonder about the possibility of advection bringing in low-VOC air, thus resulting in lower VOC than other studies. Are the field sites of Das et al. and Graus et al. much larger and thus less influenced by outside air? Any correlations with wind direction that suggest advective influences? How does the flux footprint at the measurement heights on P3 L17-18 compare with size of the field (185 x 255 m) on P3 L4? Of course, if there is an advective signal, this puts into the question the validity of the horizontal homogeneity assumption and challenges whether this is representative of north-western Europe or "ecosystem-scale."**

First of all, we apologize having misprinted the field dimension. The actual maize field is approx. 398*255 m (10.1 ha) instead of 185*255 m. The coordinates of the maize field used in all calculations were correct, though, therefore not jeopardizing the outputs of this manuscript. The field dimension will be corrected in the revised manuscript.

When talking about "lower VOC", we did not understand whether you were refering to lower VOC concentration or to lower VOC fluxes. As lower ambient VOC concentration would have resulted in higher VOC emissions from our site in comparison with other maize studies, accordingly to Niinemets et al., 2014, whereas the opposite trend was observed, we assumed that you were refering here to the possibility of underestimated VOC fluxes caused by advection processes.

Then, we are quite confident that the low measured BVOC fluxes did not result from a methodological bias induced by advective processes.

First, for footprint calculations, the variations of the displacement height (estimated as 2/3 of the maize height, the latest being estimated on a daily basis, Sec. 2.1.3) and of the mast height over the maize growing season were taken in account. Consequently, changes in footprint due to mast elevation and maize growth were taken in account. It resulted that when the maize was high (stages R1 and R2), only 7% of the data had a maize field contribution lower than 90%. This suggests that most of the time, BVOC fluxes that were measured were actually well representative of the maize field.

Second, we did not observe any influence of the wind direction that could have indicated advective processes for any compound, neither for concentration nor for flux.

Thirdly, the site is known to be poorly affected by advection processes as it is almost flat (slope of 1.2%) and located on a plateau at a large scale (no hill or mountain near the site).

Lastly, the data used to compare our exchanges with those of the other maize studies were selected to represent exchanges under warm and light conditions. Under such conditions, the turbulence is well developped, so the advective processes should be minor and thereby cannot explain the lower exchanges observed on our site in comparison with other studies.

Original text (P3 L4): *The study was carried out on a silage maize (Zea mays L., varieties Prosil and Rocket) field about 185 x 255 m [...]*

Revised text (P3 L4): *The study was carried out on a silage maize (Zea mays L., varieties Prosil and Rocket) field about 398 x 255 m [...]*

Comment n°11 (P12 L27)

**What are the "default values" and what are they based on?**

A "default" value of 2.0 µg $g^{-1}$DW $h^{-1}$ was assigned for all OVOC when there is no available emission factor for a compound and for a plant species (Karl et al., 2009). This value was taken from the "default" value assigned in the NatAir database (Steinbrecher et al., 2009) when there is not enough data available to determine experimentally an emission factor. Steinbrecher et al., 2009 indicated that they based their OVOC values on Seco et al., 2007, which reported OVOC fluxes measurements from trees ranging from 0.2 to 4.8 µg $g^{-1}h^{-1}$, and on the emission factors used in the MEGAN database (Guenther et al., 2006).

Following your question, we will specify that the "default" values are values based on the emission factors determined from other ecosystems instead of actual OVOC fluxes measurements. Note that, following the comments n°12 and 13, the text was moved after the Sec 3.5.

Original text (P12 L27): *Karl et al. (2009) noticed, however, that the SEFs given for croplands are default values because of the lack of information for those ecosystems.*

Revised text (P13 L32): *Karl et al. (2009) mentioned, however, that, because of the lack of information for those ecosystems, the SEFs for croplands are default values, i.e. values assigned by databases and up-scaling models from SEF observed on other ecosystems.*

Comment n°12 (P13 L1 to P13 L2) and

Comment n°13 (P13 L24) – answered together with Comment n°12

**Are you comfortable advising modelers to use these SEFs given all the uncertainties? Given the limited data, and the large discrepancies with the two studies cited here, I feel more data is needed to validate the SEFs found here before they are deemed a reliable representation of C4 crops in NW Europe. Instead, you might advise modelers to be wary that current SEFs may be overestimates and advise them to include that potential caveat in their studies.**

**Again, are you confident enough in your SEFs to say they "should" be used to represent C4 crop PFT?**

The best method would actually rely on a lot of BVOC data taken at similar (for repeatability) and various (for a global representativity) locations for the same plant species when assigning emission factors in up-scaling models. However, despite the huge efforts made by BVOC measurers from these last decades, there are still a lot of plant species for which few BVOC fluxes information is available. As a result, modelers have to assign emission factors relying on a few species.

Particularly, only two crop studies were considered by Stavrakou et al., 2011 when assigning methanol emission factors: Schade and Custer, 2004 and Warneke et al., 2002. The first one was done on bare soil, the other one focused on alfalfa during harvest, so when emissions should be much higher than the basal emissions, according to what has been observed on diverse crops and grasses studies. Consequently, to our opinion, the current methanol emission factors assigned for agricultural ecosystems are not representative of the actual methanol emissions from croplands, and even less for C4 crops.

On the one hand, we are confident that maize can be used alone to represent the whole C4 crops PFT in NW Europe, because it represents 99% of the total C4 crop cultivated area in that region (percentage determined by comparing the harvested surface of diverse C4 crop species, data taken from FAOSTATS for the year 2015). This may be not true for other regions where other C4 crop species are also abondant, but we did not conclude about the other regions in this manuscript.

On the other hand, we are aware that the values we propose come from one site only and need thus to be cross-validated by other studies performed on sites similar to LTO, particularly when considering the variability in BVOC exchanges rates among studies. However, as specified in the Sec. 3.1, the maize field was grown at LTO for production purposes, so the common management practices for this region were used, and the weather conditions recorded on our site were among normal conditions for Belgium. As a result, we argue that the maize field grown at LTO behaved as a common maize field grown in this region, so that the measured BVOC exchanges rates should be representative of a maize field grown under normal weather conditions in the NW region. Then, in a study performed in controlled lab conditions and focusing on maize of the same variety as the one cropped at LTO (Mozaffar et al., 2016, submitted to AE), the methanol exchange rates were of the same range as the methanol exchange rates observed in this study, thereby validating somehow the exchanges rates we proposed. Thirdly, very pragmatically, these values are currently the only ones available for maize in NW Europe. We argue thus that they should be more representative of the actual BVOC exchanges rates from maize in this region than the "default" values assigned from other crops or other ecosystems, even if their extrapolability has not been validated yet. This is why we advised the use of our factors for the C4 crop PFT in the NW region, by clearly mentioning that our results were not extrapolable to other regions or global scale.

However, following the comment n°13, we realized that we did not discuss the extrapolability of our results clearly enough, so that we did not point out that additional studies should be performed in the NW region to validate it at a regional scale, given the large discrepencies in BVOC exchange rates among maize studies. Moreover, the discussion about the representativity of the maize field grown at LTO appeared at the beginning

of the Result and Discussion part (Sec 3.1), whereas the discussion about the use of SEFs measured at LTO by up-scaling models appears at the end of the Result and Discussion part (Sec 3.5). The link between both sections was thus probably not straightforward.

We will clarify this by moving the Sec. 3.1 after the Sec. 3.5. Then, in this section, we will discuss the extrapolability of our emission rates at a regional scale, their use in up-scaling models, and adapt the section "Conclusions" accordingly. The abstract being already more moderate than the "Conclusions" section, with the word "suggest" being employed instead of "advise" or "indicate", we did not modified it. Thirdly, following the comment n°13, we will specify everywhere that we restrict the extrapolability of our results to the C4 crops PFT in the NW European region and not to all agricultural lands or the global scale.We will also restructure some paragraphs of the "Conclusion" section (P13 L23 to P14 L3) in order to make it clearer and to avoid repetitions with the revised last section of the discussion. We will however pay attention to keep the original sense of the conclusion at the exception of the use of our SEFs by up-scaling models.

Original text (P8 L23 to P8 L30 and P12 L25 to P13 L2):

[revised manuscript text omitted]

Comment n°14 (Sec 4 – Conclusions)

**Can you draw any new conclusions about the evolution of BVOC fluxes from maize fields throughout the growing season now that you have this new data set that didn't exist before? For instance, can you comment on the variability throughout the season in Figure 1 and how the "plant phenology" dependence of modeled emissions (P12 L15) captures that variability?**

We analyzed indeed the plant age effect on the methanol exchange from maize at leaf-scale and at ecosystem-scale. Results were however out of the scope of this article and were therefore not discussed. The outputs of the methanol exchange behavior along the maize growing season at leaf-scale have been very recently submitted (Mozaffar et al., 2016, submitted to AE); the ability of current BVOC up-scaling models to reproduce the methanol fluxes observed at LTO under non-standardized conditions will be detailed in a future paper.

**Technical comments**

Comment n°15 (P2 L15)

**Spell out FAOSTATS**

FAOSTATS = Food and Agriculture Organization of the United Nations Statistics Division.

Original text (P2 L15): *(FAOSTATS)*

Revised text (P2 L15): (Food and Agriculture Organization of the United Nations Statistics Division, FAOSTATS)

Comment n°16 (P8 L10)

**Spell out BBCH, and define**

BBCH is a German abbreviation that stands for Biologische Bundesanstalt, Bundessortenamt and CHemical industry (Meier 2001). It is a German scale used to identify the developmental stages of plant species. Those clarifications will be added at the first appearance of this abbreviation. Following them, the remark in P8 L 16-17 is not necessary anymore, it will thus be removed.

Original text (P8 L10): *G (germination – BBCH 00 to 14)*

Revised text (P8 L20-22): *G (germination – BBCH 00 to 14; BBCH stands for Biologische Bundesanstalt, Bundessortenamt and CHemical industry and is a decimal scale used to identify the developmental stages of plant species)*

Original text (P8 L16-17): *A detailed description of all the stages and their correspondence with BBCH codification (Meier, 2001), which is commonly used for crop phenological description, is given in Table 2.*

Revised text (P8 L27-28): *A detailed description of all the stages and their correspondence with BBCH codification (Meier, 2001) is given in Table 2.*

**Author's reply references**

Bamberger, I., Hörtnagl, L., Ruuskanen, T. M., Schnitzhofer, R., Müller, M., Graus, M., Karl, T., Wohlfahrt, G. and Hansel, A.: Deposition fluxes of terpenes over grassland, J. Geophys. Res. D Atmos., 116(14), doi:10.1029/2010JD015457, 2011.

Das, M., Kang, D., Aneja, V. P., Lonneman, W., Cook, D. R. and Wesely, M. L.: Measurements of hydrocarbon air-surface exchange rates over maize, Atmos. Environ., 37(16), 2269–2277, 2003.

Graus, M., Eller, A. S. D., Fall, R., Yuan, B., Qian, Y., Westra, P., de Gouw, J. and Warneke, C.: Biosphere-atmosphere exchange of volatile organic compounds over C4 biofuel crops, Atmos. Environ., 66, 161–168, 2013.

Guenther, A., Karl, T., Harley, P., Wiedinmyer, C., Palmer, P. I. and Geron, C.: Estimates of global terrestrial isoprene emissions using MEGAN (Model of Emissions of Gases and Aerosols from Nature), Atmos. Chem. Phys., 6(11), 3181–3210, doi:10.5194/acp-6-3181-2006, 2006.

Guenther, A. B., Jiang, X., Heald, C. L., Sakulyanontvittaya, T., Duhl, T., Emmons, L. K. and Wang, X.: The model of emissions of gases and aerosols from nature version 2.1 (MEGAN2.1): An extended and updated framework for modeling biogenic emissions, Geosci. Model Dev., 5(6), 1471–1492, doi:10.5194/acp-11-8037-2011, 2012.

Karl, M., Guenther, A., Köble, R., Leip, A. and Seufert, G.: A new European plant-specific emission inventory of biogenic volatile organic compounds for use in atmospheric transport models, Biogeosciences, 6(6), 1059–1087, doi:10.1029/2003GL017000, 2009.

Niinemets, Ü., Fares, S., Harley, P. and Jardine, K. J.: Bidirectional exchange of biogenic volatiles with vegetation: emission sources, reactions, breakdown and deposition, Plant. Cell Environ., 37(8), 1790–1809, doi:10.1111/pce.12322, 2014.

Noe, S. M., Copolovici, L., Niinemets, Ü. and Vaino, E.: Foliar limonene uptake scales positively with leaf lipid content: "Non-emitting" species absorb and release monoterpenes, Plant Biol., 10(1), 129–137, 2008.

Schade, G. W. and Custer, T. G.: OVOC emissions from agricultural soil in northern Germany during the 2003 European heat wave, Atmos. Environ., 38(36), 6105–6114, doi:http://dx.doi.org/10.1016/j.atmosenv.2004.08.017, 2004.

Seco, R., Peñuelas, J. and Filella, I.: Short-chain oxygenated VOCs: Emission and uptake by plants and atmospheric sources, sinks, and concentrations, Atmos. Environ., 41(12), 2477–2499, 2007.

Stavrakou, T., Guenther, A., Razavi, A., Clarisse, L., Clerbaux, C., Coheur, P. F., Hurtmans, D., Karagulian, F., De MaziÃre, M., Vigouroux, C., Amelynck, C., Schoon, N., Laffineur, Q., Heinesch, B., Aubinet, M., Rinsland, C. and Müller, J. F.: First space-based derivation of the global atmospheric methanol emission fluxes, Atmos. Chem. Phys., 11(10), 4873–4898, doi:10.1029/2008GL033642, 2011.

Steinbrecher, R., Smiatek, G., Köble, R., Seufert, G., Theloke, J., Hauff, K., Ciccioli, P., Vautard, R. and Curci, G.: Intra- and inter-annual variability of VOC emissions from natural and semi-natural vegetation in Europe and neighbouring countries, Atmos. Environ., 43(7), 1380–1391, 2009.

Warneke, C., Luxembourg, S. L., De Gouw, J. A., Rinne, H. J. I., Guenther, A. B. and Fall, R.: Disjunct eddy covariance measurements of oxygenated volatile organic compounds fluxes from an alfalfa field before and after cutting, J. Geophys. Res. D Atmos., 107(7-8), 1–6, 2002.